# Current Knowledge on the Transportation by Road of Cattle, including Unweaned Calves

**DOI:** 10.3390/ani13213393

**Published:** 2023-11-01

**Authors:** Kelly Buckham-Sporer, Bernadette Earley, Sonia Marti

**Affiliations:** 1Animal & Grassland Research and Innovation Centre (AGRIC), Teagasc, Grange, Dunsany, C15 PW93 Co. Meath, Ireland; 2Ruminant Production Program, IRTA, Institut de Recerca i Tecnologia Agroalimentàries, Torre Marimon, 08140 Caldes de Montbui, Barcelona, Spain

**Keywords:** transport, weaned calves, unweaned calves, animal welfare, welfare indicators

## Abstract

**Simple Summary:**

The welfare of farm animals has been, and continues to be, a key societal and political concern at a global level. Public awareness of issues surrounding the comfort and health of livestock and the transport of animals to multiple production units throughout their lifetime is also increasing. Thus, efforts to understand the impacts of common (and largely unavoidable) husbandry practices on animal health and wellbeing are appropriate, timely, and important. The aim of this review was to analyse the scientific literature available regarding the truck transportation of cattle, including unweaned calves, by road, because the infrastructure of the beef cattle industry in today’s world dictates a need for the transportation of cattle by road, including within the European Union. The buying and selling of cattle from one producer to another, the rearing of surplus calves and their movement from dairy farms to fattening farms, the finishing of weanling calves at large feedlots, and the endpoint of slaughter, all necessitate transportation. Efforts to decrease stress during the handling and transportation of unweaned calves and cattle improve welfare as well as subsequent production. Research studies have targeted such a goal, designing experiments to determine the optimum stocking density, the maximum duration of transportation, and the timing of rest stops and the determination of which components of the transportation are the most stressful. All of these examples provide incentives for implementing appropriate conditions during transportation to reach optimal animal welfare standards.

**Abstract:**

Transport conditions have the potential to alter the physiological responses of animals to the psychological or physical stress of transport. Transportation may introduce multiple physical and psychological stressors to unweaned calves and adult cattle, including noise, overcrowding, food and water deprivation, extreme temperatures, commingling with unfamiliar animals, handling by unfamiliar humans, and being placed in a novel environment upon arrival. Apart from these factors, the type of road and even driving skill may affect the welfare of animals. One of the concerns regarding cattle transport is that the handling and marketing of animals prior to a journey may lengthen the period of feed withdrawal. Furthermore, feed withdrawal can impact animal welfare through hunger and metabolic stress. Transportation is also associated with a decrease in animal performance as well as an increase in the incidence of bovine respiratory disease. It is well established that the transportation of cattle is a stressor that causes a quantifiable response; however, excessive stress during transport resulting in physiological or pathological changes can be reduced with best management practices. The objective of this review was to analyse the available scientific literature pertaining to the transport by road of cattle, including unweaned calves.

## 1. Introduction

The transport of animals can involve a combination of physical and psychological stressors, such as weaning, adverse handling during loading and unloading, commingling with unfamiliar animals, loud noises, overcrowding, food and water deprivation, extreme temperature, and the novelty of the truck or new feedlot facility, all of which can be individually stressful, let alone in combination with each other [1,2,3,4,5,6,7,8,9,10,11,12,13,14]. The time of year, ambient conditions, type, age, and breed of animals are also risks during transport [15,16]. An important concern with regard to the welfare of cattle species is traumatic injury that may occur during handling and transport [17,18,19]. This has been measured in a number of studies where cattle were being handled or transported before slaughter, and bruising on the carcass and meat quality could subsequently be assessed in cattle [19,20,21]. While Kenny and Tarrant [22] have observed that being confined on a moving vehicle is the most stressful component of transportation, others contend that loading and unloading cause the most stress to cattle [23]. What is not questioned is whether transportation can lead to negative welfare consequences [24]. Welfare consequences have been identified as being highly relevant for the welfare of cattle during transport based on severity, duration, and frequency of occurrence: group stress, handling stress, heat stress, injuries, motion stress, prolonged hunger, prolonged thirst, respiratory disorders, restriction of movement, resting problems, and sensory overstimulation [25,26,27]. A variety of hazards also leading to negative welfare consequences include inexperienced/untrained handlers, inappropriate handling, the structural deficiencies of vehicles and facilities, poor driving conditions, unfavourable microclimatic and environmental conditions, and poor husbandry practices. In addition, there is a lack of non-invasive indicators to assess the physiological state of animals during and after transport. Although cattle may be transported by ship or by rail in combination with a road journey, this review will focus on truck transportation by road, which is much more commonly practiced. Therefore, the aim of this review was to analyse the scientific literature available pertaining to the transport of adult cattle by road, including unweaned calves.

## 2. Material and Methods

The PRISMA (Preferred Reporting Items for Systematic Reviews and Meta-Analyses; and Reporting Literature Searches in Systematic Reviews) protocol was adopted in this literature review [28,29]. The literature reviewed in this manuscript were searched for on the PubMed database (https://pubmed.ncbi.nlm.nih.gov, accessed on 2 September 2022) using the terms “CALF” OR “CATTLE” OR “BULLS” OR “STEERS” AND “TRANSPORT” AND “WELFARE”. The search was performed in September 2022, January 2023, and May 2023. In addition, Google Scholar was also used to include more peer-review publications and grey literature using more specific terms of search as “CALF” OR “CATTLE” OR “BULLS” OR “STEERS” AND “ROAD TRANSPORT” AND “WELFARE” OR “HEALTH” OR “STRESS” OR “BEHAVIOUR”. Other peer-reviewed manuscripts, books, institutional reports, government publications, and PhD theses with English as the publication language were included to fill the literature gaps not found using the terms described above (Figure 1). Year of publication was not an exclusion criterium.

## 3. Evaluation of Transportation Stress

### 3.1. Measurements of Stress during Transportation

As stress occurs when an animal’s homeostasis is disrupted, the stress response consists of a set of physiological mechanisms is designed to return to homeostasis. Two distinct systems link the initial perception of the stressor to this response: the sympathetic adrenomedullary (SAM) axis and the hypothalamo–pituitary–adrenocorticol (HPA) axis. It has been suggested that the activation of the SAM axis may be associated with neurogenic stress in cattle, such as transportation and stunning at slaughter, while the activation of the HPA axis may be associated with perceived stressors, such as noise. When these stressors occur simultaneously, a mixed response is the result [30]. These two systems may result in different effects but are not entirely independent from each other, and crosstalk is involved in their regulation. For example, glucocorticoids may stimulate the synthesis of catecholamines in the adrenal medulla and catecholamines stimulate the release of ACTH from the anterior pituitary, which, in turn, stimulates the adrenal cortex to secrete glucocorticoids [31]. In addition, glucocorticoids themselves also act to dampen the HPA response when exposure to stress has subsided through the negative feedback regulation of continued ACTH release by the anterior pituitary [31,32]. The endocrine activity of the stress response is commonly measured as an indicator of stress. It is generally agreed upon that the activation of the HPA axis signals a stress response; therefore, the analysis of circulating glucocorticoids, cortisol being the most predominant, has been measured in many cattle stress studies, including lameness [33], transportation [34,35,36,37,38,39,40,41,42], branding [43], castration and ovariectomy [44,45,46,47,48], weaning [49], regrouping and relocation [50], and restraint [51]. Less frequently, indicators of the sympathetic adrenomedullary axis activation may be monitored. The measurement of the circulating catecholamines epinephrine and norepinephrine is considered a marker of very acute stress [52] and has also been utilised in bovine stress studies [43,49]. Finally, elevations in heart rate have been determined as a physiological marker of stress in cattle [43].

The measurements of stress during transport include physiological and behavioural measurements. Circulating cortisol as an indicator of the HPA axis activation is the most commonly used measurement, and increases have been observed in nearly all transportation studies of cattle as compared to pre-transportation concentrations or those obtained from non-transported counterparts [22,36,37,53,54,55,56,57,58,59]. The highest levels observed were 51.0 ng/mL after 4 h transport in Holstein steers [56] and 9.1 ng/mL after two periods of 9 h transport in 10-day-old unweaned calves [60]. However, it was reported that cortisol peaks at the beginning of the journey and this may indicate that the handling during loading is the most stressful moment of the whole transportation process [60,61,62]. A decrease in glucocorticoid receptors (GR) and β-adrenergic receptors (βAR; a catecholamine receptor) expression in lymphocytes has been observed, and the measurement of these receptors has been suggested as a more reliable indicator of stress than measurement of their corresponding stress hormones [63]. Indicators of the activation of the SAM axis are seldom used, although increases in plasma epinephrine and norepinephrine have been observed in transported calves [30,64,65]. Heart rate has also been measured as an indicator of stress during transportation. Kenny and Tarrant observed elevated heart rates in bulls while confined on a stationary truck and while on a moving truck [22]; this result is corroborated by others [66,67]. Respiratory rate has been investigated; however, no significant changes due to the stress of transportation were found [23]. 

Various changes in cattle behaviour different from those observed in a familiar environment are also indicative of stress during transportation. Transportation increased urination, defecation, and salivation in young calves and urination in mature bulls [22,55]. The frequency of social interactions such as butts, mock fights, or pushes decreased by 92%, 87%, and 91%, respectively, during transportation as compared to a resting state during the stationary time during transport or in familiar pens [22,57]. Exploratory behaviour (licking or smelling the environment) or sexual behaviour (mounting) did not increase the frequency in cattle in their resting state in comparison with the frequency observed during re-penning or confinement to a stationary truck [22]. Weaned or mature cattle prefer to stand when being transported, in an orientation that is parallel or perpendicular to the direction of the truck’s movement [67], and a high stocking density may limit their ability to change position [22], whereas unweaned calves prefer to lie down when being transported [68]. At the beginning of a journey, increased restlessness was observed: cattle were more active and spent more time standing and participating in increased social interactions [50]. As the journey time lengthened (>16 h; [57]), increased lying behaviour has been observed, either towards the end of the journey [49,61] or in a stationary vehicle [55]. Instances of ruminating, which would suggest a relaxed state in cattle, also decrease [69]. A limitation of behavioural measurements is that they may be somewhat subjective, as “normal” versus “abnormal” behaviours may be perceived differently between human observers, and it has been suggested that more research in this area is necessary. Animals that exhibited avoidance behaviours when first introduced to a potentially stressful handling procedure may become habituated and cooperate voluntarily after several occurrences [51]. However, animals under chronic stress may fail to exhibit behaviours that would be considered indicative of a stress response, such as attempting to escape. In extreme circumstances, they may adopt a “learned helplessness” over time to even the most adverse stimuli, such as electric shocks, and fail to respond at all [70]. Another reaction to chronic stress that may be misconstrued by human observers is simply an absence of normal behaviours. Animals under stress may cease to exhibit exploratory or grooming behaviours or seeking food [70]. Genetic components and previous experience cause great animal-to-animal variability with respect to the behavioural measurement of stress [70], making it difficult to determine an animal’s response to stress by its behaviour alone.

#### 3.1.1. Metabolic Changes Observed during Transportation Stress

Markers of altered protein, energy, and mineral metabolism, as well as rumen and intestinal function, have all been investigated during the transportation stress of cattle. An alteration in protein metabolism is evidenced by changes in circulating total protein, albumin, and urea, which are usually increased [35,54,55,57,58,71]; for example, Kent and Ewbank observed an increase of 7% of total protein in 1- to 3-month-old calves [55], and Tarrant et al. observed an increase of 13% in 600 kg cattle sent to slaughter [57]. Altered energy metabolism may be marked by increases in blood glucose [22,55,57,58,59], lactate dehydrogenase, glutamic pyruvic transaminase, and glutamic oxaloacetic transaminase [54]. Also, while β-hydroxybutyrate (BHB) increases between 37% to 42% in weaned calves 24 h after transport [35], in unweaned calves, this decreased between 55% and 69%, depending on the pre-transport diet [72,73]. An increase in energy metabolism is a hallmark of the stress response as the body prepares to react to a potentially dangerous situation [74]. Changes in the mineral metabolism of calcium, copper, iron, magnesium, inorganic phosphorus, potassium, and zinc were not found in weaned calves [54] or following transportation in calves of one to three weeks old [62]. An increase in blood pressure, could cause a phosphorus efflux increase, causing renal damage to the glomerulus. Studies conducted in unweaned calves showed that transport stress also increased the concentration of Cr-EDTA, an indicator of intestinal permeability, of between 58% to 64% compared with non-transported cattle [72,73], demonstrating that transport impairs gastrointestinal functionality due to the long hours of fasting.

#### 3.1.2. Changes in Performance Variables Observed during Transportation Stress

Changes in growth performance and feed intake have been investigated following transportation. Weight loss (of up to 11% total body weight) has been observed in many transportation studies (on journey durations of up to 31 h), which is attributed to the loss of gut fill, urination, dehydration, and hours of fasting [35,53,54,57,58,75,76]. When the time that cattle spend in markets, assembly centres, or control posts is considered part of the transportation process, the diet that cattle are fed during this time and prior to the journey also play important roles in the subsequent weight loss during transport [72,73].

In addition, transportation to the slaughter plant can affect carcass yields and meat quality. The loss of liveweight during transportation directly results in decreased hot carcass weights, especially at high stocking densities [57,58,59]. Plasma creatine kinase (CK) activity is often monitored as an indicator of muscle breakdown and bruising and is frequently elevated [35,54,59] by as much as 818% in a long-distance transportation [57]. Dark cutting beef is unacceptable because it is visually unappealing to consumers and high pH allows spoilage bacteria to grow, rapidly reducing its shelf life [77]. When the pH of meat remains elevated, the physical state of the proteins are above the iso-electric point, proteins retain more water, and therefore cause the muscle fibres to be more tightly packed. Consequently, the meat is becomes dark in colour because its surface does not scatter light to the same extent as the more open surface of meat with a lower ultimate pH [78]. The dark cutting condition in beef is a direct consequence of reduced muscle glycogen concentration due to exhaustion [79,80] as a result of social regrouping, usually at an abattoir [81,82]. This condition is further exaggerated by the inadequate handling of animals from the time they leave the farm to the point at which they are slaughtered [83].

Furthermore, fasting and physical stress during transportation prematurely depletes the muscle glycogen that is necessary for conversion to lactic acid and subsequent pH decrease in meat after slaughter. High meat pH (>5.8) has been observed after transport to slaughter and is associated with reduced shelf-life and the incidence of “dark cutting” or “dark, firm, and dry” (DFD) meat [57,58].

#### 3.1.3. Immunological and Inflammatory Changes Observed during Transportation Stress

Alterations in calf immunity are of great importance following transportation stress as these alterations are associated with the increased incidence and severity of respiratory disease [34]. Many measures of immunological changes relate to immune cell numbers in the blood and immune cell function. Most studies observe a leucocytosis that is marked by neutrophilia, which may occur in conjunction with a decrease in the number of other cells (lymphopaenia, eosinopaenia) [34,56,57]. On a related note, haematocrit levels are elevated during transportation in association with higher erythrocyte counts in the circulation [55,57].

Other measures include the functions of cells involved in innate immunity. Bovine alveolar macrophages, isolated from bronchoalveolar lavage (BAL) fluid, have a reduced respiratory burst function after 4 h of transportation [84]. The respiratory burst function is necessary to produce reactive oxygen species that are toxic to phagocytosed pathogens and these results may represent impaired lung defence. In contrast, enhanced respiratory burst activity has been found in the neutrophils of transported calves [56]. Yagi et al. described the decreased apoptosis of neutrophils in combination with increased migratory capacity in dairy cows after 4 h of transportation, supporting the potential enhancement of immune function [37].

Additional observations include differences in the adaptive immune response. A decrease in lymphocyte blastogenesis or cytokine production in response to an antigen has been observed [34,36,56]. Dixit et al. observed that lymphocytes produce the stress hormone ACTH and that long term transportation increased its production [66]. Interestingly, IgG_1_ concentrations were elevated in transported weaned calves as compared to non-transported controls [85], indicating a possible enhanced function of the B-lymphocyte subset. This possible enhancement of the components of immune function is supported by increases in natural killer (NK) cell counts and the expression of major histocompatibility complex class II (MHC-II) [86].

Another marker of an inflammatory response is the release of a group of proteins known as acute phase proteins. These are secreted by hepatocytes in response to injury, trauma, or infection and may be directly stimulated by glucocorticoids [87]. Their presence in the circulation may be an excellent biomarker of inflammation as they are readily measurable in serum or plasma and may even discriminate between acute and chronic inflammation in cattle [88]. Results in the literature concerning changes in acute phase protein concentrations during transportation stress are variable. Serum haptoglobin was elevated in 6-month-old steers transported for 2 days in negative correlation with lymphocyte function [89]. In a separate experiment transporting 9-month-old bulls at different stocking densities, plasma haptoglobin concentrations were unchanged, while plasma fibrinogen levels were reduced [35]. Plasma fibrinogen was significantly increased in a long-distance transportation to slaughter [57]. Fibrinogen, ceruloplasmin, serum amyloid-A, and α-acid glycoprotein were assayed in the plasma of transported and commingled calves and found to be increased post-transportation; however, haptoglobin concentrations were higher in non-transported calves in comparison with transported calves [53]. An additional inflammatory measure is oxidative stress. Oxidative stress is marked by an imbalance of reactive oxygen species produced by metabolic and inflammatory reactions and the antioxidants that neutralize these species. Oxidative stress can cause severe tissue damage, altered metabolism, and impaired reproduction in dairy cows [90]. Attenuated antioxidant capacity and elevated lipid peroxidation were observed in transported weaned steer calves in association with respiratory disease [91], indicating that the calves were under oxidative stress.

#### 3.1.4. Disease and Transport Stress

One of the concerns regarding cattle transport is that any resultant stress may be immunosuppressive and render the animals more susceptible to disease [92]. Across the different cattle categories, the prevalence of disease may not be high, but the duration of the respiratory disorders will often outlast the duration of the journey itself. Mormede et al. recorded that 8- to 32-day-old calves transported for 13 h by road had a lower incidence of disease than calves transported for twice that duration, including an overnight stop [93]. Irwin et al. identified transportation as a major risk factor in the aetiology of bovine respiratory disease [94,95]. Murata et al. transported 4- to 6-mo-old Holstein calves for 4 h by road and recorded a decrease in the populations of T-lymphocytes and suppression of lymphocyte blastogenic responses to mitogens [56]. Kelley et al. transported calves ranging in age from 1 to 20 d for up to 10 h, and they observed that older cattle also appear to be susceptible to transport-induced immunosuppression [96]. Blecha et al. transported steers (180 to 280 kg BW) for 10 h by road [34]; although blood neutrophil numbers were increased, lymphocyte blastogenic response was suppressed in transported animals, and this effect was independent of feed and water deprivation. Swanson and Morrow-Tesch reported that young cattle exhibit increased mortality and morbidity during transportation due to a lack of exposure to the new environment and a naïve immune system [1]. Buckam-Sporer et al. reported that transportation stress altered the expression of four neutrophil genes whose protein products are key in the regulation of inflammation (Fas, L-selectin, and matrix metallopeptidase 9 (MMP-9) and the clearance of Gram-negative bacterial infections (bactericidal/permeability-increasing protein (BPI)), but left the expression of three genes (betaglycan, A1, GRα) known to be affected in other models of bovine stress unchanged [97]. The gene expression profiles for the four affected genes suggested that transportation stress induced a state of heightened inflammatory and tissue-sterilizing potential in circulating neutrophils. These expression changes largely correlated with the neutrophil count profile, which relates to blood cortisol concentration in numerous models of bovine stress. The observed gene expression changes for Fas, MMP-9, L-selectin, and BPI suggested that transportation stress induced a state of heightened inflammatory activity in circulating neutrophils [98].

#### 3.1.5. Injuries and Transport Stress

An important concern with regard to the welfare of cattle is traumatic injury that may occur during handling and transport [99]. This has been measured in a number of studies where animals were being handled or transported before slaughter, and bruising on the carcasses was only assessed post-slaughter after the removal of the hair and skin, as these are not visible ante-mortem [100,101]. However, some authors have claimed that 91.2% of bruising occurs during lairage, and only 2.5%, 0.4%, and 0.5% occur during loading, transport, and unloading, respectively [102], while others have reported that transport-related bruising accounted for 13.6% in all animals and 63% of bruised cattle. One of the factors affecting severe bruising is the transport distance [103]. But, after the animal has adapted to the situation, time or distance is a minor problem compared to loading densities, vehicle design, road conditions, or a driver’s driving behaviour [104]. Most losses of balance during transportation that result in injury and bruising are related to driving events and occur during the cornering and braking of the truck by the driver [105]. In addition, bruising that occurs during transit reduces meat quality, and carcass bruise scores have been observed to increase linearly with stocking density [57]. In the SCAHAW report concerning the welfare of animals during transport, it was discussed whether cattle, in situations of poor driving or emergency responses, could benefit from a stocking density that provided them with mutual support [106]. However, based on the available studies, cattle seem to be at greater risk of stress and injuries at low space allowances than with high [57,107]. Garcia et al. and Mendonça et al. reported that bruising scores increased with lower space allowance (stocking densities above 401 kg/m^2^) [108,109]. Previously, Tarrant et al. found that 600 kg cattle began to lie down after 16 h of transport, but at the highest stocking density of 600 kg/m^2^, the animals could not rest because of the lack of space [57]. Although cattle prefer to stand during transport, they do lie down during long journeys [58]. Thus, preventing animals from resting after 16 h or more of transport may become an important animal welfare issue in many countries. Studies on the relationships between vehicle design, transit conditions, climatic conditions, transport time, and distance are required to obtain a better insight regarding their effects on bruising occurrence.

The response of tissue to a bruise-inducing event depends on the nature of the mechanical force applied and also on the anatomical location where the force is applied. Anderson and Horder have suggested that in beef cattle, external factors (i.e., source, transport, and handling) may be responsible for the site where bruises are located in the body of the animal, whereas animal factors, such as presence of horns, sex class, and temperament, determine the severity of bruising and may cause deeper lesions [110]. A study on skin lesions caused by transport in water buffaloes examined various forms of skin lesions, namely abrasions, lacerations, penetrating wounds, ulcerations, bleeding sores, swelling with hyperkeratosis, and scar tissue, with abrasions and lacerations being the most common skin injuries in this type of animals with frequencies of 95.3% and 57.3% [111]. The areas most affected by these types of lesions were the buttocks, hips, and back, and water buffaloes were more affected by skin lesions than *Bos indicus* [111,112]. In a recent study conducted in Italy in beef cattle, bruising was observed more often in the flanks and hips (Figure 2) in veal and young cattle (slaughtered < 12 month) and heifers (1 to 3 years old), representing the main cattle classifications at the greatest risk of bruising [19].

### 3.2. Factors Affecting the Extent of Transportation Stress

A number of varying factors may affect the extent of stress that animals experience when being transported in addition to those mentioned previously regarding stress in general. These include the age of the cattle, breed differences, previous experience of the animals regarding handling and transportation, road conditions and driver variables, stocking density, outside temperature, duration of the journey, and more [22,58,69]. The negative impact of these factors may lead to extreme stress during transportation and the subsequent incidence of bovine respiratory disease (BRD).

#### 3.2.1. Age of the Cattle during Transportation

The age of the cattle transported can have a significant effect as instances of morbidity and mortality increased in transported calves younger than 3 weeks of age [113]. Swanson and Morrow-Tesch reported that transport might be more stressful in older (>3-month-old) compared to young (<4-week-old) cattle due to reduced lying and ruminating and greater salivation, defaecation, and urination in older cattle [1]. Kent and Ewbank reported similar observations [62]. However, younger calves do not show the same HPA response as older calves, and this complicates the evaluation of transport stress in younger calves [93]. The immune components of calves are not completely functional at 2 weeks of age, and they are in the so-called “immune gap period” due to the combination of decreasing passive immunity and the absence of a mature adaptive immune system (Figure 2) [114]. A recent study by Marcato et al. found that calves transported to a veal farm at 4 weeks of age showed more advanced adaptive immunity development than calves transported at 2 weeks of age [41], even though those calves were still in the “immune gap period”. A calf’s immune system does not reach full maturity until they are ~6 months of age (Figure 3) [115,116,117]. Transport during this period increases the risk of mortality and morbidity, especially regarding bovine respiratory disease and diarrhoea: mortality rates ranged from 0.06% to 0.7% in the 24–30 h after calves’ arrival [118,119], and between 33% and 96% of young calves were treated with antimicrobial medication during the first few weeks after transport to the rearing farm [40,120]. Therefore, for the more successful transportation of unweaned calves, colostrum provision at birth is crucial. Furthermore, although increasing the transport age of unweaned calves from 2 weeks to 4 weeks of age results in more robust calves facing transport [41], the care and welfare of the calves that stay longer at their original dairy farm cannot be jeopardized as all the benefits of increasing age at transport will be impaired.

From 6 to 8 weeks of age, the active immune system may be sufficiently developed for calves to withstand the transport challenge [121]. However, 75% of the morbidity and 50% of the mortality in feedlots in United States is caused by BRD [122] with an overall incidence of BRD of 14.4% [123], mainly 21 days after arrival to feedlots [124]. Even though older cattle may be more robust from an immunological point of view to face transport challenges, consequences on health due to stressors during transport are similar among animals different ages.

#### 3.2.2. Type of Cattle

Several breed differences have been found between calves of Bos indicus and Bos taurus breeding during weaning and transportation [125], and it is generally agreed upon that cattle with genetically more excitable temperaments may remain agitated during handling procedures and transportation [51]. Cattle that are habituated to human presence and handling, and calves that are group-reared exhibit lower plasma cortisol concentrations and lower heart rates following handling and transportation than extensively reared cattle or calves reared in isolation without social experience [69,126].

#### 3.2.3. Road Conditions and Driver Variables

Road conditions are another contributing factor, and elevated heart rates have been observed in cattle on rough country roads or suburban roads with many stops and turns as compared to those being transported on highways [67]. Most losses of balance during transportation that result in injury and bruising are related to driving events and occur during the cornering and braking of the truck, thus adding the variable of the driver [22]. The stress caused by road conditions, driver variables, and the injuries associated with these also may increase the incidence of “dark cutting” meat [57,127]. González et al. reported an increase in the probability of animals becoming non-ambulatory or totally compromised during transport when drivers had less than 5 years of hauling experience [3]. Agnes et al. used a transport simulator to subject calves (75 kg BW) to transport loading or transport noise without motion [64]. The responses in plasma adrenalin, serum cortisol, and non-esterified fatty acid concentrations were compared with those of calves that underwent complete transport simulation. Adrenalin, cortisol, and non-esterified fatty acids increased for all treatments compared with pre-treatment concentrations. There were no differences in calf response to loading, noise, or simulated transport.

#### 3.2.4. Stocking Density

The term “stocking density” is used to express the weight of the animal per unit floor area (e.g., kg m^2^) but is often used in place of “stocking rate”, which describes the number of animals per unit floor area (e.g., head per m^2^). In turn, “stocking rate” may also be used to describe the inverse which, correctly, is the “space allowance”: the floor area allocated per animal (e.g., m^2^ per head) [128]. Many studies have investigated the variable of stocking density or space allowance. It has been determined that an optimum stocking density exists in which animals are not overcrowded but can still lean on each other during the cornering and braking of the truck. Increased stocking density results in increased plasma cortisol concentrations, heart rate, the occurrence and severity of injury and bruising, and decreased carcass weight [126].

When space allowance was evaluated for unweaned calves, Todd et al. observed that calves transported at a space allowance of 0.2 m^2^/calf stood more than calves at 0.4 m^2^/calf [129]; similar results were observed by Uetake et al. in calves transported at 0.25, 0.35, and 0.45 m^2^/calf, where the lying time and occurrence of turning was higher as space allowance increased [130]. Neonate calves oriented to the direction of travel for the greatest duration (40% of the time during transit) and then preferred transverse orientation (30% of the time during transit). The least-preferred orientation was diagonal and facing the tail-gate (19 and 11% of the time during transit) [1]. Grigor et al. observed that 10-day-old calves transported at 0.475 m^2^/calf oriented themselves either parallel (mean proportion of observations = 0.36) or perpendicular (mean proportion of observations = 0.37) to the direction of travel compared with calves transported at 0.375 m^2^/calf, which oriented themselves perpendicular to the direction of travel (mean proportion of observations = 0.54), whereas parallel orientations were less common (mean proportion of observations = 0.15) [60]. Although the reason for the preferred orientation is unknown, it could be related to balance, air movement, or to ensure better calf stability when lying or standing.

When space allowance was evaluated for weaned calves, Tarrant et al. observed that calves transported at densities of 200, 300, and 600 kg/m^2^ [107]; calves transported at stocking rates of 1.37, 1.22, and 1.06 m^2^/head with an average of 618 kg [57]; or calves that were transported at an density of 600 kg/m^2^ and stocking rates of 1.06 m^2^/head were subject to increased struggling and falls, resulting in more bruising observed at the slaughterhouse. The preferred standing orientation was standing across the truck at right angles to the direction of motion; however, driving conditions may cause standing calves to end up facing the direction of the cab [22].

Increased lying behaviour after a journey is observed and might be due to inadequate space for calves to lie down and rest during the journey [131]. Grigor et al. observed that the time taken for the calves to first lie down in their pens after a journey was shorter in calves that had been transported at 0.375 m^2^/calf (mean latency was 966 s) compared with calves transported at 0.475 m^2^/calf (mean latency was 2022 s) [60]. Similar results were observed by Jongman and Butler when lying time was measured at recovery in 3-to-10-day old calves, with an average weight of 38 kg, and compared calves transported at 0.2 m^2^/calf compared to 0.3 m^2^/calf [68]. There is relatively little knowledge on fatigue effects during transport, although it is a reasonable assumption that animals become tired if they are standing for a long journey. There are limited data on how tired animals become and what the associated risks are [19].

When calves are transported with low space allowances, their opportunity for movements is restricted. Jongman and Butler observed a reduction in the number of posture changes in calves transported at 0.2 m^2^/calf than at 0.3 and 0.5 m^2^/calf [68]. Uetake et al. observed an increase in the occurrence of turning when space allowance increased from 0.25 to 0.35 and to 0.45 m^2^/calf [130]. In addition, Todd et al. observed that calves transported at a space allowance of 0.2 m^2^/calf remained standing for longer than calves at 0.4 m^2^/calf and concluded that calves transported with a lower space allowance experienced greater muscular fatigue [129]. Plasma CK activity increased more than two-fold in calves transported at 0.2 m^2^/calf compared with calves transported at 0.3 m^2^/calf or 0.5 m^2^/calf. Additionally, elevated CK activity was observed in calves transported at 0.3 m^2^/calf compared with 0.5 m^2^/calf, which may have been due to the risk of injury and fatigue at low stocking densities [68]. In older calves, as they stand during transport, changes in posture are not common; however, calves changed position at an average of 8.5 events/group/h when the truck was stationary and 6.9 events/group/h when the truck was moving [22].

Regulation (EC) 1/2005 established a range of space for animals based on different animal categories and weights [132]. Petherick and Phillips have proposed the use of an allometric equation to establish the physical space requirements that increase with increasing BW [133]. The space needed can be calculated using the formula: A = k × (BW)^2/3^; where A is the floor area covered by the calf; k is a constant value that depends on calf position; and BW is individual body weight. This formula is useful for providing standardized space allowance recommendations across all cattle regardless of their weight [5,24]. A k value of at least 0.02 has been proposed to provide sufficient space for a standing position [133]; a k value of 0.027 provides sufficient space to allow all animals to lie down in a sternal position [133]; a k value of 0.033 to provide sufficient space to allow all the animals to lie down and stand up again simultaneously [134]. All these k values are designed for animals in pens, and further research should be carried out to evaluate space requirements during transport (Table 1).

#### 3.2.5. Transport Duration

Cattle transported over long distances may become physically fatigued from attempting to stand and may also suffer from dehydration and hunger. During a 31 h journey, rest stops did not prove effective as cattle appeared too agitated to drink water or to lie down [58]. However, even short journeys may result in poor animal welfare if the transportation conditions are very poor, i.e., poor ventilation, extreme temperatures, overstocking, etc. [126]. All of these factors need to be considered when ameliorating the impact of transport stress.

Maximum journey duration is one of the major concerns of animal welfare during transport, especially for unweaned calves. A positive relationship between transport distance or duration and metabolic changes or mortality has been observed, specifically when distances were greater than 400–500 km [118,135]. Boulton et al. estimated that the odds ratio for young calf mortality increased by a factor of 1.45 for each additional hour that calves were transported [136]. In the latter study, bobby calves were a minimum age of 4 days old and the maximum journey times were lower than 8 h. Therefore, as suggested by Roadknight et al. reducing transport distances and duration may reduce mortality and improve the welfare of calves during and after transportation [135]. Wilson et al. observed 58.8% and 33.9% of unweaned calves with poor or very poor eye and nose scores when calves were exposed to at least two different transports and one temporary residence on arrival at the rearing facility due to dust, engine fumes, or air velocity; in addition, 49.5% and 60% of the calves also had poor or very poor scores for front and rear hoofs and feet on arrival [137]. Longer journey durations may increase the risk of animals becoming ill due to the longer exposure time to extreme temperatures, longer time off feed and water, or difficulties in resting, as well as impaired environmental conditions, all of which may reduce the capacity of the immune system of the animals and increase susceptibility to disease pathogens. Mormede et al. observed that calves less than 32 days old transported for long and short journeys had mean pathology ratings of 3.78 and 2.47, respectively [93]. Longer periods of feed restriction and fasting in calves has also been reported to affect gut permeability. The serum concentration of Cr-EDTA was reported to be greater in calves that were feed-restricted and fasted for 19 h compared to 9 h [72,73]. Consequently, with increasing journey durations, the free passage of molecules, microorganisms, and other pathogens from the intestinal lumen to the bloodstream may render calves more susceptible to disease.

##### Rest Periods

In many countries, there are regulations to provide animals with some hours of rest in resting stations or at control posts during long-distance transport. During this time, cattle are provided with feed and water and a period of rest to recover from the first transport. However, resting time increases the total time of the journey to the final destination, the stress due to the additional loading and unloading, the exposure to new environments, and the risk of exposure to disease (Tarrant and Grandin, 2000). Recently, the effect of resting time during different journey times on different types of cattle and their origins have been investigated [138,139]. In general, resting times of < 4 h do not benefit weaned calves when performance was evaluated [138,140]. In addition, resting times of 8 h and 12 h improved arrival body weight, although shrink was greater when resting for 12 h [138]. Marti et al. (2017) found that recently weaned calves did not avoid short- and long-term stress after 20 h of transport with rest periods of longer than 10 h [141]. The successful feeding of weaned calves at rest stations depends on different factors. Meléndez et al. observed that non-conditioned weaned calves had difficulty in benefitting from feeding during resting periods as these animals are not familiar with the feed bunks [139]. Ross et al. reported that doubling the feeding space at the rest stations increased the mean proportion of calves eating and reduced interruptions during feeding time among cattle [142]. Access to water in the resting facilities is of great importance, as the time when cattle have the greatest need to access a drinker is after unloading and this requirement decreases with time [143]. Finally, for a beneficial rest, bedding is crucial, as the latency and duration of lying behaviour is conditioned by it [144,145]. The benefits of unloading cattle at a rest station for feed, water, and rest to improve their welfare does not always result in a straightforward positive outcome. Despite some performance data, other welfare indicators are inconsistent [138,139] and longer periods in rest stops are risk factors that increase the abundance of BRD-associated genera that affect the upper respiratory tracts of cattle and calves [146].

#### 3.2.6. Hot and Cold Weather

The scientific literature on the relationships among in-transit microclimates is speculative [7], is based on environmental/ambient conditions [2,147,148] and not the conditions of cattle inside transporters, or is not applicable to transport conditions due to differences in types of transporter and space allowances among studies [6]. Based on the current literature, 25 °C is proposed as the upper critical temperature for cattle as studies have shown that respiration rate and the rate of sweating increased at 25 °C, and rectal temperature increased after 26 °C (Figure 4) [149,150]. At temperatures higher than that, unweaned calves and cattle suffer heat stress.

Unweaned calves, due to their low fat reserves and in some cases failure of passive immunity transfer, may suffer cold stress. The thermoneutral zone in young calves ranges from 15 °C to 25 °C [150,151,152], although metabolic changes start to show when the air temperature falls below 5 °C. Heat stress in cattle can be defined as the animal’s inability to dissipate sufficient heat to maintain homeothermy, which is mainly due to high air temperature but is intensified by high humidity, thermal radiation, low air movement, and metabolic heat. High ambient temperature accompanied by high air humidity causes discomfort and increases the stress level in animals, which in turn impairs their physiological and metabolic activities [153,154,155,156]. The inability to regulate body temperature results in energy reserves being diverted to thermoregulatory processes such as decreased heat production during heat stress, thereby reducing energy reserves for growth performance, lactation, or pregnancy.

Maintaining homeothermy in a hot environment essentially depends on an animal’s ability to balance thermogenesis and heat dissipation [157,158]. Increased respiration rates eliminate a greater volume of carbon dioxide, which can lead to a disturbance in the acid–base balance [159]. Excessive salivation usually accompanies increased respiration rate [160,161,162]. Norris et al. found that the respiratory rate of cattle increased by 5.7 breaths per minute for every 1 °C ambient temperature greater than 25 °C [163].

Rectal temperature (RT) is the most common indicator of body temperature and has been widely used as a physiological indicator of heat stress [164,165,166]. The thermal comfort zone for cattle is a body temperature (BT) of between 36.7 °C and 39.1 °C [167]. The optimum rectal temperature in cattle is between 38.0 °C and 39.2 °C [164,168]. It is also possible to detect changes in the temperature of the skin and extremities using infrared thermography (IRT) as the temperature of those regions is dependent on peripheral blood flow [169,170]. This technique, for example, has enabled the non-invasive analysis of ocular surface temperatures as well as those of other body parts when assessing transport stress in calves [171] and buffaloes [172]. The precise determination of external body surfaces using IRT depends on the effective minimisation of factors, such as skin and hair colour, the emissive properties of the skin, sudden movements, health status, the time of feeding, and external factors, including ambient temperature, relative humidity, sunlight, wind speed, distance between camera lens and the measuring object, and angle of camera [173].

#### 3.2.7. Fitness for Transport

One of the main concerns is the lack of protocols and agreements to guide when calves are fit for transport. When calves, independently of age, are unfit for transport, their capacity to withstand the challenges faced during transport is reduced, thus further compromising their health and welfare [174,175]. Feed withdrawal before transport can impact animal welfare through hunger and metabolic stress [152]. Previous clinical conditions that cause pain to calves are aggravated during transport, increasing their suffering [175]. In young calves, alterations in their immune functions represent additional potential impacts of transport [34,176]. For this reason, the preparation of fitness for transport should be taken in account at the farm of origin. An important consideration is that unweaned calves are more difficult to move than older cattle since they do not show natural herding behaviour [177], thereby increasing the risk of poor handling [135] and injury, resulting in discomfort and pain [135]. Gregory et al. reported that slips and falls were the main problems during loading [178], and Bravo et al. found that slips and vocalizations were more common during unloading, whereas slips and turning around occurred more frequently during loading [179]. It has also been reported that group-housed calves are more difficult to load than individually kept calves [69,180,181]. While Albright et al. found that group-housed calves were more explorative when loaded onto a vehicle [180], and Trunkfield and Broom reported more balking and turning in group-housed calves than in individually housed calves [69]. Lensink et al. observed that group-housed calves stopped more often during loading, as indicated by the number of pushes performed by the handlers [181]. The lack of aggressive behaviour by unweaned calves should make the mixing easier; however, mixing animals from different origins may increase the risk of spread of diseases among them. Marcato et al. investigated effects of pre-transport diet (rearing milk vs. electrolytes), transport duration (6 vs. 18 h), and transport conditions (open truck vs. conditioned truck) on the metabolic and physiological variables of young calves (mean weight: 45.3 kg (range: 36.0–56.4 kg)) upon arrival at the veal farm [40]. Blood parameters were analysed until week 5 post-transport to study the recovery time of calves after arrival. The authors found that the increase in NEFA and BHB concentrations between pre- and just post-transport (T0) was less pronounced in calves transported for 6 h (746.1 µmol/L and 0.38 mmol/L, respectively) than in calves transported for 18 h (850.6 µmol/L and 0.50 mmol/L). Overall, the recovery rate of calves at the veal farm seemed rapid; all blood parameters returned to (below) pre-transport values within 48 h post-transport. The authors concluded that feeding milk before short-term transport helps young veal calves cope with transport, whereas this is not the case during long-term transport. There are also studies focusing on body weight (and not age) of calves to evaluate the risk of transport, and it has been reported that calf weight has a high impact on morbidity and mortality [38].

## 4. Concerns about Transportation Stress

### 4.1. Economic Concerns

The economic impact of transportation is of great concern to all involved in the cattle industry. Some aspects to consider are the cost of preventative measures and treatment of shipping fever, meat quality, and yield concerns and the occurrence of “downer cattle” and mortality [182,183]. Respiratory diseases in cattle, including the occurrence of shipping fever following transportation, account for the greatest proportion of cattle losses in the world [184]. While not all transport of cattle results in BRDs, the temporal proximity between various stressors in feedlot cattle, such as weaning, transport, housing, and dietary change results in immunosuppression, thus predisposing the respiratory tract to colonization by infectious pathogens [92,185]. According to the 2011 National Animal Health Monitoring System (NAHMS) feedlot report, 96.9% and 16.2% of feedlots and cattle were affected by shipping fever, respectively [186]. Studies on BRD have reported the impact of lung lesions identified at slaughter on the growth performance of feedlot cattle; compared to animals without lung lesions during the late fattening period, those with severe lung lesions had a reduction in growth rate of 5.3% (1.67 vs. 1.58 kg/d) [187] and 16.7% (1.8 vs. 1.5 kg/d) [188]. It is estimated that preventative and therapeutic costs account for approximately 7% of the total production cost in USA cattle [182]. With respect to meat quality, injuries incurred during transit may necessitate the trimming and disposal of bruised areas of muscle, resulting in major economic losses [126,127]. Liveweight losses resulting in reduced hot carcass weights decrease profits. Transportation stress is also associated with the incidence of “dark cutting” or “dark, firm, and dry” (DFD) meat, which is highly undesirable for consumers [57,127]. In addition, the periods of pre-slaughter fasting due to transportation can be difficult to control, yet there is ample evidence that extended periods of feed withdrawal can result in bodyweight loss and reductions in meat quality in animals that are transported to slaughter [124,189]. Cattle with non-ambulatory status (“downer cattle”) that have been badly injured on the transporter may not be accepted at slaughter plants, resulting in, again, a profit loss for the producer [183]. Finally, mortality can occur during transportation, which is rare in finished cattle but much higher in young calves [113,183], leading to direct economic losses.

### 4.2. Welfare Concern

In addition to economic concerns, there is a growing concern for animal welfare during stressful handling procedures, such as transportation. Welfare is a complex concept and many definitions exist. It has been proposed that animal welfare refers to the individual’s ability to adapt to the environment, based on the presumption that animals can suffer and feel pain and discomfort; therefore, animal welfare can be measured scientifically and can range from very good to very bad [190,191]. Animal welfare has also been defined based on the affective state of the animals [192] or using a multidimensional approach, and encompasses the natural life, affective state, and biological functioning of animals [193]. According to this concept, a good state of welfare can only be achieved when elements of natural life (such as fresh air and the ability to express natural and innate behaviours), affective needs (free from intense negative states such as pain and fear and being able of feel positive emotions), and the normal biological functioning (growth, health) of the animals are assured [193]. Appropriate ways to assess the state of animals are needed to avoid mistreatment and to give animals “a life worth living” [194,195]. According to Broom, animal welfare can be measured by physiological and behavioural indicators of pleasure; the extent of behavioural aversion shown [196]; physiological attempts to cope; immunosuppression; disease prevalence; behavioural attempts to cope; behaviour pathology; brain changes; body damage prevalence; reduced ability to grow or breed; and reduced life expectancy. It is suggested by Broom and Johnson that quantitative methods should be used to measure welfare, such as physiological indicators (e.g., heart rate, adrenaline, cortisol concentrations, and others) [197]. The most widely used physiological indicators of impaired welfare are related to stress response, distress, disease, and pain [198]. Both the effort involved in an animal’s attempt to cope and the possibility that its coping attempts fail are indicators of poor welfare [190]. While stress is not necessarily synonymous with poor welfare, many of the negative effects of transportation stress are consistent with those that are indicators of poor welfare. Some examples of this are failure to exhibit normal behaviours, weight loss and decreased growth rate, increased incidence of injury and disease, and possible mortality. Grandin has observed that efforts to decrease stress during the handling and transportation of cattle and pigs improve welfare as well as subsequent production [199]. Many studies carried out within the European Union aim at such a goal, designing experiments to determine the optimum stocking density, the maximum duration of transportation, and the timing of rest stops and determining which components of the transportation are the most stressful to cattle [22,35,57,58,59,71]. 

### 4.3. Societal Concerns

A recent paper by Bachelard discusses transport stress and although regulations exist, there is still the need for further improvement [200]. Animal welfare is concerned with the wellbeing of the animal and complements the objectives of assurance schemes that demonstrate the production of safe food to consumers and food chain stakeholders [201,202]. Consumers perceive the need to increase the level of welfare in farm animals, and certain industry practices such as livestock transport [203] are losing their social licence to operate [204] despite the fact that the level of knowledge about farming and animal welfare issues may still be relatively low [202]. Interactions with animal welfare science, engagement with stakeholders, the transparency of animal welfare reporting, reliance on public relations, and the role of the media are some of the factors that may contribute towards losing the social licence to operate [205]. In the EU, producers are required to meet public standards in animal health and welfare, as outlined in EU and national legislation. More recently, this has been supplemented with private standards in animal health and welfare, as part of quality assurance programmes, which are generally aligned to socially important issues, related to the environment (for example, greenhouse gas emissions), the product (quality and safety), or the animals (health and welfare). Vigors et al. suggest that the consideration of both scientific and societal perspectives points to an approach to welfare that accounts for both positive and negative experiences, prioritises them (e.g., by seeing positive experiences as dependent on basic animal needs being fulfilled), and considers the balance of positives and negatives over the lifetime of the animals [206].

### 4.4. Limitations and Perspectives

In this review, the key factors affecting animal welfare during transportation (e.g., heat stress, animal type, stocking density, the duration of transport, rest stops, pre-transport conditions) were highlighted. Throughout the scientific literature, exposure to high or low temperatures is considered a challenge for animals during transport. Although transportation has been studied over the years, there are still gaps in our knowledge that warrant further investigation. In the recent scientific opinions on the welfare of animals during transport, EFSA also came to this conclusion, and selected “heat stress” as a highly important welfare consequence across the animal species that are typically farmed and transported in the EU [24,207,208,209].

Considering that the current expectation is an increase in warmer and more variable climates in the future, it can also be assumed that this challenge to animal welfare during transport will also increase. The microclimate surrounding animals during transport is complex, and is a function of both internal and external factors, including stocking density, climate, ventilation strategy, heat and moisture generated by the animals, and vehicle design and driver quality. Some of these factors should be studied together to understand how cattle of different age and/or climate origin face the thermal challenge of heat stress during transport.

The lack of feed and the difficulties of accessing water during transport is another key factor in the welfare of cattle during transportation. In this review, it was highlighted that animals being unloaded in a resting facility is not always the best solution due to the risk of increased stress and respiratory problems. However, no research has examined the unloading of unweaned calves at a control post for milk feeding. Factors regarding this scenario, at least for unweaned calves when transport durations are longer than 8 h, may necessitate the feeding of those calves inside the truck. Today, there are mechanical possibilities for feeding unweaned calves with milk replacer whilst they are inside the truck during a 1 h rest that follows an 8 h transport period. However, it is unknown how much milk those calves could drink and the effect this may have when calves digest the milk while the truck is in motion. Feeding cattle during a long journey also warrants further investigation, regarding whether they are unloaded and fed at a control post or offered feed on the transporter during a journey (amount and concentration, type of feed, time, etc.).

One of the main animal health outcomes associated with transportation is BRD. Cattle are social animals and establish social bonds and hierarchy among themselves. The social hierarchy of cattle determines the priority of access to resources. The abrupt breakage of the social bond or hierarchy through regrouping and relocating may lead to social stress and an animal may respond with abnormal behaviour and consequently alter its biology by evoking significant changes in its normal body metabolism and the neuro-immuno-endocrine system. Most of the cattle transported are still marketed through live auction markets and are often placed in assembly centres to manage batches of similar type and size pre-transport. All of these multiple transports, environments, and commingling of animals are part of the journey, extending the transport times to the final destination and thus also increasing the risk of disease and mortality. This process could be improved when source farms are large enough to sell directly to a rearing farm without having to go through live auction markets or assembly centres. However, this may not always be feasible considering that the average dairy farm in Europe has 55 cows [210]. However, as observed by Meléndez et al. the conditioning of calves, in particular unweaned calves, before leaving the farm of origin needs to be further investigated in order to reduce the negative impact of transport. Furthermore, fitness for transport requires more specific and detailed protocols that take into consideration borderline cases, where the line between fit and unfit is blurred, when deciding whether animals can be transported either to another farm or to a slaughterhouse. Farmers have already expressed concerns regarding the lack of knowledge to decide whether cattle are fit for transport, and this leads to frustration regarding the lack of uniformity and predictability of outcomes when cattle are inspected at slaughterhouses [211]. For example, a study reported that 20% of calves had swollen navels when presented for sale in auction markets [212]. Calves with swollen navels are often transported under “slightly injured or ill and transport would not cause additional suffering” [132]; however, often they have to be euthanized at the destination farm a few days after arrival due to their deteriorated condition. Discount payments for those animals that have to be euthanized or die after arrival at rearing farms is not the solution as the animals should not have been transported in the first place. Consequently, there is a need for more specific and detailed protocols on “fitness for transport” so as to avoid unnecessary stress to animals.

## 5. Conclusions

The current infrastructure of cattle production around the world, including within the European Union, dictates the use of truck transportation. As this scenario is unlikely to change in the future, the health and welfare of transported cattle remains a concern. It is recognised that transport is an essential process that can potentially result in stress, injury, and disease in cattle and unweaned calves, and continued research is necessary to elucidate the mechanisms behind these negative effects and to devise methods to reduce their occurrence. The stress associated with the transportation of cattle based on behavioural, physiological, and production responses has been well studied. However, little work has been carried out on deciding the degree to which deviation from normal values is acceptable and how transportation can render cattle, including unweaned calves, more susceptible to disease. Moreover, it is of interest to examine the effect of transportation on the stress adaptability of cattle that were previously exposed to acute or chronic stress stimuli (e.g., housing with high stocking densities). We suggest that further research is therefore needed to identify the factors or combination of factors during transport with the greatest negative impacts on animals to ensure the continual improvement of the welfare of cattle, including unweaned calves, undergoing transport.

## Figures and Tables

**Figure 1 animals-13-03393-f001:**
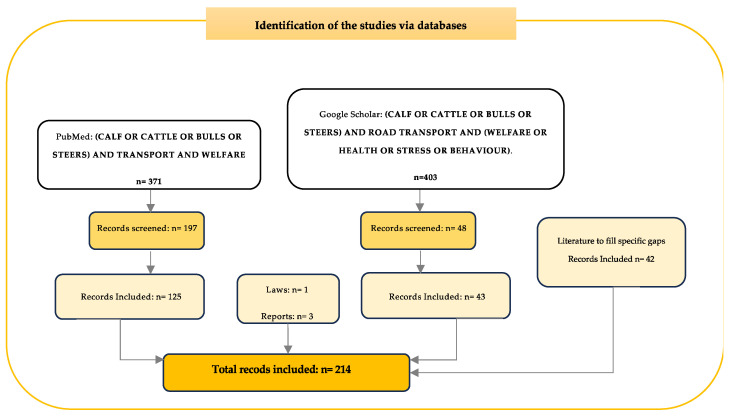
PRISMA flow diagram of the literature search for the review process.

**Figure 2 animals-13-03393-f002:**
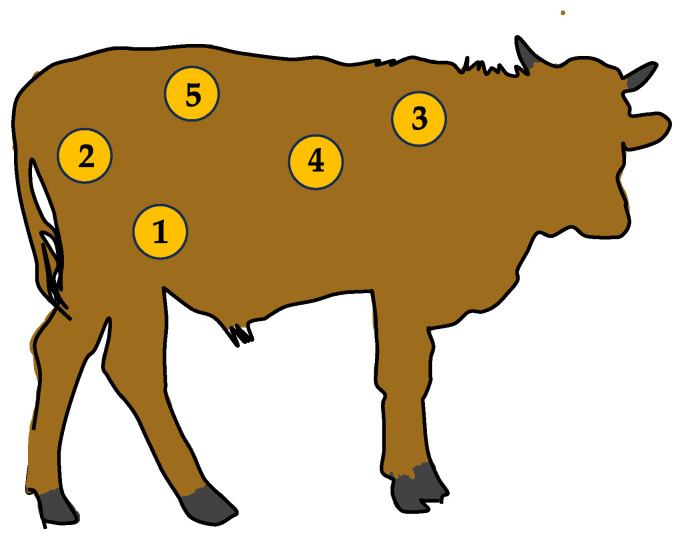
Frequency of bruises related to transport according to body regions identified by Zanardi et al. 1—flank (39.5%); 2—buttock (36.0%); 3—front region (23.8%); 4—ribs (12.8%); and 5—loins (13.7%) [19]. Modified from José-Pérez et al. [112].

**Figure 3 animals-13-03393-f003:**
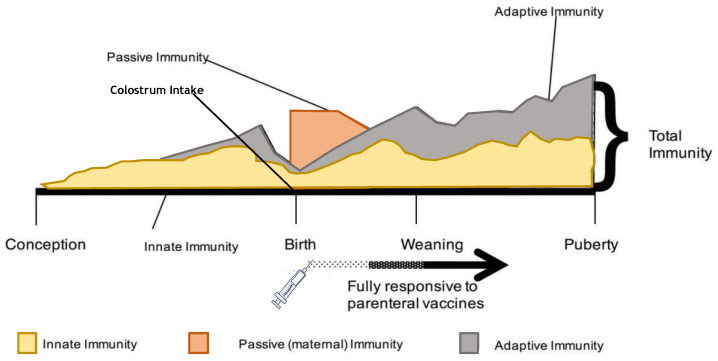
Development of the immune response from conception to puberty in calves (adapted from: Morein et al., 2002, and Chase, 2018 [115,117]).

**Figure 4 animals-13-03393-f004:**
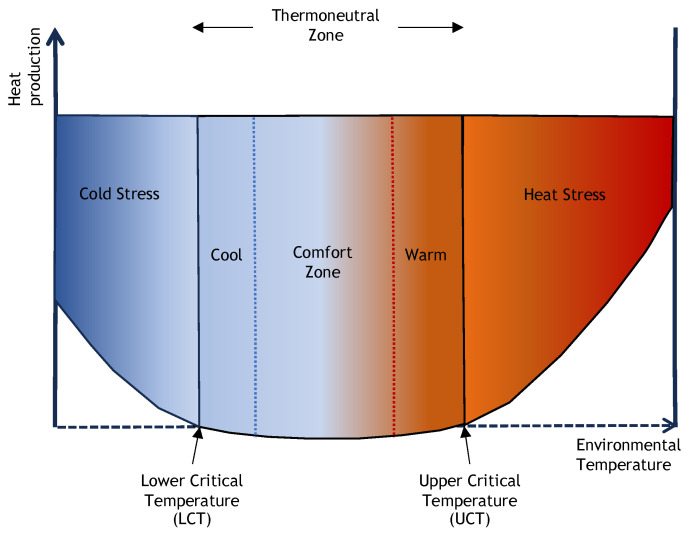
Schematic representation of thermal zones as a function of the environmental temperature. The thermoneutral zone for animals is the range of temperatures below which an animal feels cold and has to divert extra energy for heat production and above which it feels hot and reduces its heat production by decreasing feed intake (adapted from EFSA Scientific Opinion [24]).

**Table 1 animals-13-03393-t001:** Physical space requirements based on the RE (CE) 1/2005 ^1^, and k as a constant value that depends on calf position: k = 0.02 for standing; k = 0.027 for lying in a sternal position; and k = 0.033 for allowing cattle to change positions from lying down and standing up.

Category	Weight, kg	Area m^2^/calf ^1^	Area m^2^/calf, k = 0.02	Area m^2^/calf, k = 0.027	Area m^2^/calf, k = 0.033
Small calves	50	0.30–0.40	0.27	0.37	0.45
Medium calves	110	0.40–0.70	0.46	0.62	0.76
Heavy calves	200	0.70–0.95	0.68	0.92	1.13
Medium size cattle	325	0.95–1.30	0.95	1.28	1.56
Heavy cattle	550	1.30–1.60	1.34	1.81	2.22
Very heavy cattle	>700	>1.60	>1.58	>2.13	>2.60

## Data Availability

Not applicable.

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
