# Peer review of "Current Knowledge on the Transportation by Road of Cattle, including Unweaned Calves"

_animals, 2023, doi:10.3390/ani13213393_

Round 1

Reviewer 1 Report (Previous Reviewer 1)

Comments and Suggestions for Authors

Authors:

-The article is well-written and offers valuable contributions worthy of publication.

 -One of my primary concerns is the limited discussion of thermal imbalances in cattle when addressing transportation stress. This article is pivotal in explaining the significance of thermal imbalances in different body regions. I strongly recommend its inclusion in the review: https://doi.org/10.3389/fvets.2023.1184577.

 -Another critical aspect for the review would be to emphasize the injuries that livestock experience during transportation, focusing on the primary regions regularly affected. I suggest reviewing this article, which is essential to this topic: http://dx.doi.org/10.31893/jabb.22016.

 -It would be important to include a brief section on 'Limitations and Perspectives' that enables us to comprehend the current dilemmas and areas requiring further study in this field before delving into the conclusions.

 -The conclusions are somewhat limited and do not align closely with the central objective of the review. Once the authors make these minimal adjustments, I recommend that the article be published.

 Respectfully

Author Response

Reviewer 1

Comments and Suggestions for Authors

Authors:

The article is well-written and offers valuable contributions worthy of publication.

  • One of my primary concerns is the limited discussion of thermal imbalances in cattle when addressing transportation stress. This article is pivotal in explaining the significance of thermal imbalances in different body regions. I strongly recommend its inclusion in the review: https://doi.org/10.3389/fvets.2023.1184577.

Authors’ responses

Thank you for your comment; We have included the information on thermal imbalances in different body regions and referenced IRT transport studies at lines 594-604

It is also possible to detect changes in temperature of the skin and extremities using infrared thermography (IRT) as the temperature of those regions is dependent on peripheral blood flow [Stewart et al., 2007; McManus et al., 2016;]. This technique, for example, has permitted the non-invasive analysis of ocular surface temperatures for transport stress assessment, and other body parts including calves [Lei et al., 2023] and buffalos [Rodríguez-González et al. 2023) allowing for a correlation with the animal’s internal body temperature. The precise determination of external body surfaces using IRT depends on the effective minimisation of factors, such as, skin and hair colour, emissive properties of the skin, sudden movements, health status, time of feeding, and external factors such as ambient temperature, relative humidity, sunlight, wind speed, distance between camera lens and the measuring object, and angle of camera [Idris et al., 2022].

  • McManus C., Tanure C.B., Peripolli V., Seixas L., Fischer V., Gabbi A.M., Menegassi S.R.O., Stumpf M.T., Kolling G.J., Dias E., et al. Infrared Thermography in Animal Production: An Overview. Comput. Electron. Agric. 2016;123:10–16. doi: 10.1016/j.compag.2016.01.027. [CrossRef] [Google Scholar]
  • Stewart M., Webster J.R., Verkerk G.A., Schaefer A.L., Colyn J.J., Stafford K.J. Non-Invasive Measurement of Stress in Dairy Cows Using Infrared Thermography. Physiol. Behav. 2007;92:520–525. doi: 10.1016/j.physbeh.2007.04.034. [PubMed] [CrossRef] [Google Scholar]
  • Lei MC, Félix L, Cardoso R, Monteiro SM, Silva S, Venâncio C. Non-Invasive Biomarkers in Saliva and Eye Infrared Thermography to Assess the Stress Response of Calves during Transport. Animals (Basel). 2023 Jul 14;13(14):2311. doi: 10.3390/ani13142311. PMID: 37508087; PMCID: PMC10376388.
  • Rodríguez-González D, Guerrero Legarreta I, Cruz-Monterrosa RG, Napolitano F, Titto CG, Abd El-Aziz AH, Hernández-Avalos I, Casas-Alvarado A, Domínguez-Oliva A, Mota-Rojas D. 2023. Assessment of thermal changes in water buffalo mobilized from the paddock and transported by short journeys. Front Vet Sci. 2023 May 12;10:1184577. doi: 10.3389/fvets.2023.1184577. PMID: 37252398; PMCID: PMC10217363.
  • Idris M, Uddin J, Sullivan M, McNeill DM, Phillips CJC. Non-Invasive Physiological Indicators of Heat Stress in Cattle. Animals (Basel). 2021 Jan 2;11(1):71. doi: 10.3390/ani11010071. PMID: 33401687; PMCID: PMC7824675.

  • Another critical aspect for the review would be to emphasize the injuries that livestock experience during transportation, focusing on the primary regions regularly affected. I suggest reviewing this article, which is essential to this topic: http://dx.doi.org/10.31893/jabb.22016.

Authors’ response

Thank you for the comment. We have reviewed the paper on transport of buffaloes and injuries experienced as a consequence of transport, and added more information citing articles focused in cattle specifically. In addition, a new figure was created (Figure 4) based on the suggested article.

Additional modifications are made at the following lines:

At lines 516-521 we have added:

Wilson et al. observed 58.8% and 33.9% of unweaned calves with poor or very poor eye score and nose score when calves are exposed to at least two different transport and one temporary residence on arrival at the rearing facility due to dust, engine fumes or air velocity; also 49.5% and 60% of the calves had poor or very poor scores of the front and rear hoof and foot on arrival as well (Wilson et al. 2000).

At lines 303-310:

This has been measured in a number of studies where animals were being handled or transported before slaughter, and bruising on the carcass could subsequently be assessed after removing the skin at the slaughterhouse due to the hair coat and thick skin of cattle (Wigham et al. 2018 and Strappini et al. 2009). However, some authors have described that 91.2% of the bruising occur during lairage, and only 2.5%, 0.4% and 0.5% occur during loading, transport and unloading respectively (Strappini et al. 2009) while others reported that transport-related bruising accounted for 13.6% of the total of animals and 63% of the bruised cattle. One of the factors affecting sever bruising is the transport distance (Losada-Espinosa et al. 2021).

At lines 312-315:

Most losses of balance during transportation that result in injury and bruising are related to driving events and occur during cornering and braking of the truck, adding the variable of the driver (Kenny et al. 1987).

At lines 336-345:

A study in skin lesions in water buffaloes due to transport have classified the skin lesions by abrasions, lacerations, penetrating wounds, ulcerations, bleeding sores, swelling with hyperkeratosis, and scar tissue being the abrasions and lacerations with a frequency of 95.3% and 57.3% the most common skin injuries in this type of animals (Alam et al. 2010). The areas more affected by these types of lesions were manly the buttocks, hips and back and water buffaloes were more affected by skin lesions than Bos indicus (Alam et al. 2010; José-Pérez et al. 2022). In a recent study conducted in Italy in beef cattle, bruising was observed more often in the flank and hips (Figure 4) with veal and young cattle (slaughtered < 12 month) and heifers (1 to 3 years old) as the main cattle classification with greater bruising (Zanardi et al. 2022).

At lines 660-670:

With respect to meat quality (Schwartzkopf-Genswein et al., 2012), injuries incurred during transit may necessitate the trimming and disposal of bruised areas of muscle, resulting in major economic losses (Knowles, 1999; Tarrant, 1990). Liveweight losses resulting in reduced hot carcass weights decrease profits. Transportation stress is also associated with the incidence of “dark cutting” or “dark, firm, and dry” (DFD) meat, which are highly undesirable by consumers [Knowles, 1999; Tarrant et al., 1992)]. Cattle with non-ambulatory status (“downer cattle”) that have been badly injured on the transporter may not be accepted at slaughter plants, resulting in, again, a loss profit to the producer [Speer et al., 2001].

  • Alam, M.R.; Gregory, N.G.; Jabbar, M.A.; Uddin, M.S.; Kibria, A.S.M.G.; Silva‐Fletcher, A. Skin Injuries Identified in Cattle and Water Buffaloes at Livestock Markets in Bangladesh. Veterinary Record 2010, 167, 415–419, doi:10.1136/vr.c3301.
  • José-Pérez, N, Mota-Rojas, D, Ghezzi, MD, Rosmini, MR, Mora-Medina, P, Bertoni, A, Rodríguez-González, D., Domínguez-Oliva, A., Guerrero Legarreta., I. 2022. Effects of transport on water buffaloes (Bubalus bubalis): factors associated with the frequency of skin injuries and meat quality. J Anim Behav Biometeorol. (2022) 10:1–9. doi: 10.31893/jabb.22016
  • Minka N. S., Ayo J. O. 2017. Effects of different road conditions on rectal temperature, behaviour and traumatic injuries during transportation of different crosses of temperate/tropical breeds of heifers. Animal Production Science 58, 2321-2328.
  • Miranda-de La Lama, G.C., Villarroel, M. and María, G.A., 2014. Livestock transport from the perspective of the pre-slaughter logistic chain: a review. Meat Science, 98(1), pp.9-20.
  • Knowles, T. G. 1999. A review of the road transport of cattle. Vet. Rec. 144: 197-201.
  • Speer, N. C., G. Slack, and E. Troyer. 2001. Economic factors associated with livestock transportation. J. Anim. Sci. 79(E suppl): E166-E170.
  • Strappini, A.C.; Metz, J.H.M.; Gallo, C.B.; Kemp, B. Origin and Assessment of Bruises in Beef Cattle at Slaughter. Animal 2009, 3, 728–736, doi:10.1017/S1751731109004091.
  • Strappini, A.C.; Metz, J.H.M.; Gallo, C.; Frankena, K.; Vargas, R.; de Freslon, I.; Kemp, B. Bruises in Culled Cows: When, Where and How Are They Inflicted? Animal 2013, 7, 485–491, doi:10.1017/S1751731112001863.
  • Schwartzkopf-Genswein, K.; Faucitano, L.; Dadgar, S.; Shand, P.; González, L.; Crowe, T. Road transport of cattle, swine and poultry in North America and its impact on animal welfare, carcass and meat quality: A review. Meat Sci. 2012, 92, 227–243. [Google Scholar] [CrossRef] [PubMed]
  • Tarrant, P. V. 1990. Transportation of cattle by road. Appl. Anim. Behav. Sci. 28: 153-170.
  • Tarrant, P. V., F. J. Kenny, D. Harrington, and M. Murphy. 1992. Long distance transportation of steers to slaughter: effect of stocking density on physiology, behaviour, and carcass quality. Livest. Prod. Sci. 30: 223-238.
  • Wilson, L.L.; Smith, J.L.; Smith, D.L.; Swanson, D.L.; Drake, T.R.; Wolfgang, D.R.; Wheeler, E.F. Characteristics of Veal Calves upon Arrival, at 28 and 84 Days, and at End of the Production Cycle. J Dairy Sci2000, 83, 843–854, doi:10.3168/jds.S0022-0302(00)74948-4.
  • Wigham, E.E.; Butterworth, A.; Wotton, S. Assessing Cattle Welfare at Slaughter – Why Is It Important and What Challenges Are Faced? Meat Sci 2018, 145, 171–177, doi:10.1016/j.meatsci.2018.06.010.
  • Zanardi, E.; De Luca, S.; Alborali, G.L.; Ianieri, A.; Varrà, M.O.; Romeo, C.; Ghidini, S. Relationship between Bruises on Carcasses of Beef Cattle and Transport-Related Factors. Animals 2022, 12, 1997, doi:10.3390/ani12151997.

  • It would be important to include a brief section on 'Limitations and Perspectives' that enables us to comprehend the current dilemmas and areas requiring further study in this field before delving into the conclusions.

Authors’ responses

We have inserted an new section to address 'Limitations and Perspectives'

Lines:731-789

In this review, the key factors affecting animal welfare during transportation (e.g., heat stress, animal type, stocking density, duration of transport, rest stops, pre-transport conditions) were highlighted. Throughout the scientific literature, exposure to high or low temperatures is considered a challenge for animals during transport. Although transportation has been studied over the years, they are still limitations in knowledge that needs attention. In the recent scientific opinions on the welfare of animals during transport, EFSA also came to this conclusion, and selected ‘heat stress’ as a highly important welfare consequence across the animal species typically farmed and transported in the EU [24,206–208].

Considering that the current expectation is a warmer and more variable climate into the future, it can also be assumed that this challenge to animal welfare during transport will also accentuate. The microclimate of animals during transport is complex being a function of internal and external factors including, stocking density, climate, ventilation strategy, heat and moisture generated by the animals and vehicle design and driver quality. Some of these factors should be studied together to understand how cattle of different age and/or climate of origin faces the thermal challenge of heat stress during transport.

The lack of feed and the difficulties to have access to water during transport is another key factor on the welfare of cattle during transportation. In this review, it was highlighted that animals being unloaded in an resting facility is not always the best solution for the risk of increased stress and respiratory problem, however no research has examined the unloading of unweaned calves in an control post for milk feeding. Perspectives in this area, at least for unweaned calves if transport durations are longer than 8 h, may necessitate the feeding of those animals inside the truck. Nowadays there exist technical options to feed milk replacer while unweaned calves are inside the truck during the 1 h rest that follows an 8 h transport. However, it is unknown how much milk those calves may drink and the effect this may have when calves digest the milk when the truck is in motion. Feeding cattle during a long journey also warrants further investigation, whether they are unloaded and fed in a control post, or if they are offered feed on the transporter during a journey (amount and concentration, type of feed, time,…).

 One of the main animal health outcomes associated with transportation is BRD. Cattle are social animals and establish social bonds and hierarchy among themselves. The social hierarchy of cattle determines the priority of access to resources. The abrupt breakage of the social bond or hierarchy through regrouping and relocating may lead to social stress and an animal may respond with abnormal behaviour and consequently alter its biology by evoking significant changes in its normal body metabolism and neuro-immuno-endocrine system. Most of the cattle transported are still marketed through live auction markets and are often placed in assembly centres to manage batches of similar type and size pre-transport. All of these multiple transports, environments and commingling are part of the journey extending the transport times to the final destination and the risk of disease and mortality. This process could be improved when the source farms are large enough to sell directly to the rearing farm without having to go through live auction markets or assembly centres. However, this may not always be feasible considering that the average dairy farm in Europe has 55 cows [209]. However, as observed by Meléndez et al. conditioning of calves, in particular unweaned calves, before leaving the farm of origin needs to be further investigated so as to reduce the negative impact of transport.

Furthermore, fitness for transport requires more specific and detailed protocols that takes into consideration borderline cases, where the line between fit and unfit is blurred, when deciding if animals in that state can be transported either to another farm, or trans-ported to a slaughterhouse. Farmers already have expressed concerns on the lack of knowledge to decide if cattle are fit for transport, and this leads to frustration on the lack of uniformity and predictability of outcomes when cattle are inspected at the slaughterhouse [215]. For example, a study reported that 20% of calves had swollen navels when presented for sale in auction markets [216]; those calves with swollen navels are often transported under “slightly injured or ill and transport would not cause additional suffering” [135] however, often they have to be euthanized at the destination farm a few days after arrival due to their deteriorated condition. Discount payments for those animals that have to be euthanized or die after arrival to the rearing farms is not the solution as the animals should not have been transported in the first place. Consequently there is a need for more specific and detailed protocols on “fitness for transport” so as to avoid unnecessary stress to animals.

  • The conclusions are somewhat limited and do not align closely with the central objective of the review. Once the authors make these minimal adjustments, I recommend that the article be published.

Authors’ responses

Conclusion have been updated in Lines 797-806.

Reviewer 2 Report (Previous Reviewer 2)

Comments and Suggestions for Authors

Dear authors,

Animal transportation is an important issue and I am always happy to see work done in this area. 

However, I have two concerns regarding this manuscript:

1) Your overall focus is not clear to me. The title reads "adult cattle and un-weaned calves", yet in line 65 of the Introduction the aim is "weaned and un-weaned calves". I think this is a problem throughout the text. Do you wish to compare calves to adult cattle in terms of transport stress, or un-weaned to weaned calves? Many times you just write "calves" and it is not clear whether you are referring to un-weaned or weaned calves. It would be very helpful for the reader if this was clarified.

2) I recognize two of your names from the recent EFSA Scientific Opinion on Welfare of Cattle During Transport. I find the EFSA report quite thorough and it also has a section on un-weaned calves. I am curious to learn why you think this manuscript needs to be published as well? It is even the same figures you use. Please explain.

Author Response

Reviewer 2

Comments and Suggestions for Authors

Dear authors,

Animal transportation is an important issue and I am always happy to see work done in this area. 

However, I have two concerns regarding this manuscript:

1) Your overall focus is not clear to me. The title reads "adult cattle and un-weaned calves", yet in line 65 of the Introduction the aim is "weaned and un-weaned calves". I think this is a problem throughout the text. Do you wish to compare calves to adult cattle in terms of transport stress, or un-weaned to weaned calves? Many times you just write "calves" and it is not clear whether you are referring to un-weaned or weaned calves. It would be very helpful for the reader if this was clarified.

Authors’ responses

Apologies for our lack of clarity.

The review is not a comparison but a review of the literature concerning the transport of cattle, including unweaned calves. We have revised the aim to read “The study objective was to analyze the available scientific literature pertaining to the transport by road of adult cattle, including unweaned calves.

2) I recognize two of your names from the recent EFSA Scientific Opinion on Welfare of Cattle During Transport. I find the EFSA report quite thorough and it also has a section on un-weaned calves. I am curious to learn why you think this manuscript needs to be published as well? It is even the same figures you use. Please explain.

Authors’ responses

The authors were invited by ANIMALS, in July, 2022, to contribute a review paper to the Special Issue of ANIMALS, on transport after finishing the EFSA report. We have added some sections in this manuscript that goes beyond the Scientific Opinion. Figures also have been edited.

Reviewer 3 Report (New Reviewer)

Comments and Suggestions for Authors

The paper, titled “Current knowledge on the transportation of adult cattle and un-2 weaned calves by road“ addresses an important and timely topic. I found the subject matter of the article fascinating and read the manuscript with great interest. The paper aligns well with the scope of the journal. However, I believe that in its current form, it has several shortcomings.

The aim of this review paper is to comprehensively analyze the existing scientific literature concerning road transportation of weaned and un-weaned calves, a critical aspect of the beef cattle industry. As the demand for cattle transportation continues, it's essential to ensure the welfare of these animals during transit. The paper highlights the various stressors and factors affecting the well-being of transported cattle, from overcrowding to food deprivation and road conditions. By examining the literature, this paper contributes to our understanding of the challenges associated with cattle transportation and identifies potential areas for improvement in animal welfare during this process. Its strength lies in its comprehensive analysis of a topic of significant importance to both the industry and animal welfare advocates.

This paper is highly relevant to the scope of the journal. It delves into the critical aspect of animal welfare during road transportation of weaned and un-weaned calves, an issue of growing concern in the livestock industry. The paper provides a thorough analysis of the scientific literature on this topic, contributing valuable insights that align with the journal's focus on animal science and welfare.

While the paper attempts to review the literature, it could benefit from a more structured and organized approach. Categorizing the studies reviewed based on key factors affecting animal welfare during transportation (e.g., stocking density, duration of transport, rest stops) could make the review more reader-friendly and help identify research gaps.

The paper's structure and organization could be improved for better readability. Clear section headings and subheadings, along with a logical flow of information, would enhance the paper's structure.

Specific comments:

I suggest rewriting the simple summary. According to the author's guidelines, this section should summarize and contextualize your paper within the existing literature in your field. It should be written without technical language or nonstandard acronyms, with the goal of conveying the meaning and importance of this research to non-experts.

Introduction:

The introduction must to be enlarged with another concept, the animal transport influence also the quality of the production, such as the meet, as reporte in recente papers, please consider in adding this informaiton and citing these valuable researches (10.3390/ani10122386 and 10.3390/ani10060945).

The paper lacks a clearly defined hypothesis or specific research objectives. Formulating a research question or hypothesis would provide a more structured focus for the review.

Methods:

Since this is a literature review, there are no direct methodological inaccuracies. However, the paper should clearly state the criteria used to select and include studies in the review to ensure transparency and rigor in the review process.

I suggest reporting more details and references regarding the PRISMA procedure you follewed, as example and for citing you can report: 10.3390/educsci12080573.

Discussion:

While the paper touches on various stressors during transportation, it doesn't delve into the economic aspects. Including information on the economic implications of improved animal welfare during transport, such as reduced veterinary costs or improved meat quality, could enhance the paper's relevance.

The paper briefly mentions the potential impacts of stress during transportation on animal performance and health. Expanding on these implications and suggesting areas for future research could provide valuable insights.

While the paper identifies the stressors associated with cattle transportation, it could do more to pinpoint specific research gaps in the existing literature. This would guide future research efforts.

Conclusion:

The paper concludes without summarizing the key findings or providing a synthesis of the reviewed literature. A well-structured conclusion that highlights the main takeaways from the review would be beneficial.

In summary, while the paper provides a valuable review of the literature on calf transportation, addressing the above points would enhance its clarity, focus, and overall contribution to the field of animal welfare during transit.

Author Response

Reviewer 3

Comments and Suggestions for Authors

The paper, titled “Current knowledge on the transportation of adult cattle and unweaned calves by road“ addresses an important and timely topic. I found the subject matter of the article fascinating and read the manuscript with great interest. The paper aligns well with the scope of the journal. However, I believe that in its current form, it has several shortcomings.

The aim of this review paper is to comprehensively analyze the existing scientific literature concerning road transportation of weaned and un-weaned calves, a critical aspect of the beef cattle industry. As the demand for cattle transportation continues, it's essential to ensure the welfare of these animals during transit. The paper highlights the various stressors and factors affecting the well-being of transported cattle, from overcrowding to food deprivation and road conditions. By examining the literature, this paper contributes to our understanding of the challenges associated with cattle transportation and identifies potential areas for improvement in animal welfare during this process. Its strength lies in its comprehensive analysis of a topic of significant importance to both the industry and animal welfare advocates.

This paper is highly relevant to the scope of the journal. It delves into the critical aspect of animal welfare during road transportation of weaned and un-weaned calves, an issue of growing concern in the livestock industry. The paper provides a thorough analysis of the scientific literature on this topic, contributing valuable insights that align with the journal's focus on animal science and welfare.

While the paper attempts to review the literature, it could benefit from a more structured and organized approach. Categorizing the studies reviewed based on key factors affecting animal welfare during transportation (e.g., stocking density, duration of transport, rest stops) could make the review more reader-friendly and help identify research gaps.

  • The paper's structure and organization could be improved for better readability. Clear section headings and subheadings, along with a logical flow of information, would enhance the paper's structure.

Authors’ responses

We have changed the structure and heading and sub-heading of some sections. We hope that the reviewer finds it more reader-friendly. Also we have inserted text under perspectives and limitations that might be of help to identify research gaps.

Specific comments:

  • I suggest rewriting the simple summary. According to the author's guidelines, this section should summarize and contextualize your paper within the existing literature in your field. It should be written without technical language or nonstandard acronyms, with the goal of conveying the meaning and importance of this research to non-experts.

Authors’ responses

Thank you for your comment.

We have revised the summary.  We are restricted on word count and have avoided the use of technical language and non-standard acronyms.

Introduction:

  • The introduction must to be enlarged with another concept, the animal transport influence also the quality of the production, such as the meet, as reporte in recente papers, please consider in adding this informaiton and citing these valuable researches

(10.3390/ani10122386  Sardi et al. (2020) and 10.3390/ani10060945 Sardi et al. (2020))

Authors’ responses

Thank you for the comment. We have included text with reference to transport and meat quality. We have focused solely on bovine as it is the subject of the review. We are not dismissing the research on pigs, and we recognise the importance of the work cited on pigs, but in the context of the title of the review we have confined the references to the bovine species. We trust that this is acceptable. Also as requested for other reviewer the section of injuries and meat quality have been completed with more information.

Lines 52-56

An important concern with regard to the welfare of cattle species is traumatic injury that may occur during handling and transport [Grandin, 1994; Huertas et al., 2010]. This has been measured in a number of studies where cattle were being handled or transported before slaughter, and bruising on the carcass and meat quality could subsequently be assessed [Hultgren et al., 2022; Carrasco-García et al., 2022].

  • Grandin, T. 1994. Farm animal welfare during handling, transport and slaughter.  Am. Vet. Med. Assoc. 204:372.
  • Hultgren, J., Arvidsson Segerkvist, K., Berg, C., Karlsson, A.H., Öhgren, C., Algers, B.2022.  Preslaughter stress and beef quality in relation to slaughter transport of cattle, Livestock Science, Volume 264, 105073, ISSN 1871-1413, https://doi.org/10.1016/j.livsci.2022.105073.
  • (https://www.sciencedirect.com/science/article/pii/S1871141322002475)
  • Huertas, S.M., Gil, A.D., Piaggio, S.M., van Eerdenburg, F.J.C.M. 2010. Transportation of beef cattle to slaughter- houses and how this relates to animal welfare and carcass bruising in an extensive production system. Welf., 19 (2010), pp. 281-285
  • Carrasco-García A, Pardío-Sedas V, León-Banda G, et al. 2020. Effect of stress during slaughter on carcass characteristics and meat quality in tropical beef cattle. Anim Biosci 2020;33(10):1656-1665.
  •  

  • The paper lacks a clearly defined hypothesis or specific research objectives. Formulating a research question or hypothesis would provide a more structured focus for the review.

Authors’ responses

Thank you for the comment. We think it may be somewhat difficult to formalise a hypothesis for the review paper. We have taken your comment into consideration and we have rephrased the objective of the review in Lines 71-72.

Methods:

  • Since this is a literature review, there are no direct methodological inaccuracies. However, the paper should clearly state the criteria used to select and include studies in the review to ensure transparency and rigor in the review process.

I suggest reporting more details and references regarding the PRISMA procedure you followed, as example and for citing you can report: 10.3390/educsci12080573.

Authors’ responses

The methodology has been updated, adding references to the protocol used and more criteria were listed.

  • Hutton, B.; Salanti, G.; Caldwell, D.M.; Chaimani, A.; Schmid, C.H.; Cameron, C.; Ioannidis, J.P.A.; Straus, S.; Thorlund, K.; Jansen, J.P.; et al. The PRISMA Extension Statement for Reporting of Systematic Reviews Incorporating Network Meta-Analyses of Health Care Interventions: Checklist and Explanations. Ann Intern Med2015, 162, 777–784, doi:10.7326/M14-2385.
  • Rethlefsen, M.L.; Kirtley, S.; Waffenschmidt, S.; Ayala, A.P.; Moher, D.; Page, M.J.; Koffel, J.B.; Blunt, H.; Brigham, T.; Chang, S.; et al. PRISMA-S: An Extension to the PRISMA Statement for Reporting Literature Searches in Systematic Reviews. Syst Rev 2021, 10, 39, doi:10.1186/s13643-020-01542-z.

Discussion:

  • While the paper touches on various stressors during transportation, it doesn't delve into the economic aspects. Including information on the economic implications of improved animal welfare during transport, such as reduced veterinary costs or improved meat quality, could enhance the paper's relevance.

Authors’ responses

We have expanded the section on economic aspects

Lines 645-672:

The economic impact of transportation is of great concern to all involved in the cattle industry. Some aspects to consider are the cost of preventative measures and treatment of shipping fever, meat quality and yield concerns, the occurrence of “downer cattle,” and mortality [183,184]. Respiratory diseases in cattle, including the occurrence of shipping fever following transportation, account for the greatest proportion of cattle losses in the world [185]. While not all transport of cattle results in BRD, the temporal proximity between various stressors in feedlot cattle, such as weaning, transport, housing and dietary change results in immunosuppression, thus predisposing the respiratory tract to colonization by infectious pathogens [92,186]. According to the 2011 National Animal Health Monitoring System (NAHMS) feedlot report, 96.9% and 16.2% of feedlots and cattle were affected by shipping fever, respectively [187].  Studies on BRD have reported the impact of lung lesions identified at slaughter on growth performance of feedlot cattle; compared to animals without lung lesions during the late fattening period, those with severe lung lesions had a reduction of growth rate by 5.3% (1.67 v. 1.58 kg/d) [188], and 16.7% (1.8 v. 1.5 kg/d) [189]. It is estimated that preventative and therapeutic costs account for approximately 7% of the total production cost in USA cattle [183]. With respect to meat quality, injuries incurred during transit may necessitate the trimming and disposal of bruised areas of muscle, resulting in major economic losses [127,190]. Liveweight losses resulting in reduced hot carcass weights decrease profits. Transportation stress is also associated with the incidence of “dark cutting” or “dark, firm, and dry” (DFD) meat, which are highly undesirable by consumers [127,128]

  • Blakebrough-Hall, C., McMeniman, J.P., González, L.A. An evaluation of the economic effects of bovine respiratory disease on animal performance, carcass traits, and economic outcomes in feedlot cattle defined using four BRD diagnosis methods. J Anim Sci 98(2)doi: 10.1093/jas/skaa005
  • Earley, B., Buckham Sporer, K., Gupta, S.  Invited review: Relationship between cattle transport, immunity and respiratory disease. Animal 11(3):486-492. doi: 10.1017/s1751731116001622
  • Griffin, D. 1997. Bovine Respiratory Disease update: Economic impact associated with respiratory disease in beef cattle. Vet. Clin. North Am.: Food Anim. Pract. 13: 367-377.
  • Lekeux, P. 1995. Bovine Respiratory Disease Complex: An European perspective. Bovine Pract. 29: 71-75.
  • Lynch, E. M., McGee, M., Doyle, S., Earley, B. 2011. Effect of post-weaning management practices on physiological and immunological responses of weaned beef calves. Ir J Agric Food Res 50:161-174.
  • Thompson, P. N., A. Stone, and W. A. Schultheiss. 2006. Use of treatment records and lung lesion scoring to estimate the effect of respiratory disease on growth during early and late finishing periods in South African feedlot cattle. J Anim Sci 84(2):488-498. doi: 10.2527/2006.8424
  • Feedlot 2011. Part IV: Health and Health Management on U.S. Feedlots with a Capacity of 1000 or More Head; NAHMS, USDA: Fort Collins, CO, USA, 2013.

  • The paper briefly mentions the potential impacts of stress during transportation on animal performance and health. Expanding on these implications and suggesting areas for future research could provide valuable insights. While the paper identifies the stressors associated with cattle transportation, it could do more to pinpoint specific research gaps in the existing literature. This would guide future research efforts.

Authors’ responses

We have added a section of limitations and perspectives to cover the reviewer suggestion.

Lines:731-789

In this review, the key factors affecting animal welfare during transportation (e.g., heat stress, animal type, stocking density, duration of transport, rest stops, pre-transport conditions) were highlighted. Throughout the scientific literature, exposure to high or low temperatures is considered a challenge for animals during transport. Although transportation has been studied over the years, they are still limitations in knowledge that needs attention. In the recent scientific opinions on the welfare of animals during transport, EFSA also came to this conclusion, and selected ‘heat stress’ as a highly important welfare consequence across the animal species typically farmed and transported in the EU [24,206–208].

Considering that the current expectation is a warmer and more variable climate into the future, it can also be assumed that this challenge to animal welfare during transport will also accentuate. The microclimate of animals during transport is complex being a function of internal and external factors including, stocking density, climate, ventilation strategy, heat and moisture generated by the animals and vehicle design and driver quality. Some of these factors should be studied together to understand how cattle of different age and/or climate of origin faces the thermal challenge of heat stress during transport.

The lack of feed and the difficulties to have access to water during transport is another key factor on the welfare of cattle during transportation. In this review, it was highlighted that animals being unloaded in an resting facility is not always the best solution for the risk of increased stress and respiratory problem, however no research has examined the unloading of unweaned calves in an control post for milk feeding. Perspectives in this area, at least for unweaned calves if transport durations are longer than 8 h, may necessitate the feeding of those animals inside the truck. Nowadays there exist technical options to feed milk replacer while unweaned calves are inside the truck during the 1 h rest that follows an 8 h transport. However, it is unknown how much milk those calves may drink and the effect this may have when calves digest the milk when the truck is in motion. Feeding cattle during a long journey also warrants further investigation, whether they are unloaded and fed in a control post, or if they are offered feed on the transporter during a journey (amount and concentration, type of feed, time,…).

 One of the main animal health outcomes associated with transportation is BRD. Cattle are social animals and establish social bonds and hierarchy among themselves. The social hierarchy of cattle determines the priority of access to resources. The abrupt breakage of the social bond or hierarchy through regrouping and relocating may lead to social stress and an animal may respond with abnormal behaviour and consequently alter its biology by evoking significant changes in its normal body metabolism and neuro-immuno-endocrine system. Most of the cattle transported are still marketed through live auction markets and are often placed in assembly centres to manage batches of similar type and size pre-transport. All of these multiple transports, environments and commingling are part of the journey extending the transport times to the final destination and the risk of disease and mortality. This process could be improved when the source farms are large enough to sell directly to the rearing farm without having to go through live auction markets or assembly centres. However, this may not always be feasible considering that the average dairy farm in Europe has 55 cows [209]. However, as observed by Meléndez et al. conditioning of calves, in particular unweaned calves, before leaving the farm of origin needs to be further investigated so as to reduce the negative impact of transport.

Furthermore, fitness for transport requires more specific and detailed protocols that takes into consideration borderline cases, where the line between fit and unfit is blurred, when deciding if animals in that state can be transported either to another farm, or trans-ported to a slaughterhouse. Farmers already have expressed concerns on the lack of knowledge to decide if cattle are fit for transport, and this leads to frustration on the lack of uniformity and predictability of outcomes when cattle are inspected at the slaughterhouse [215]. For example, a study reported that 20% of calves had swollen navels when presented for sale in auction markets [216]; those calves with swollen navels are often transported under “slightly injured or ill and transport would not cause additional suffering” [135] however, often they have to be euthanized at the destination farm a few days after arrival due to their deteriorated condition. Discount payments for those animals that have to be euthanized or die after arrival to the rearing farms is not the solution as the animals should not have been transported in the first place. Consequently there is a need for more specific and detailed protocols on “fitness for transport” so as to avoid unnecessary stress to animals.

Conclusion:

  • The paper concludes without summarizing the key findings or providing a synthesis of the reviewed literature. A well-structured conclusion that highlights the main takeaways from the review would be beneficial.

Authors’ responses

As also requested reviewer 1, conclusion have been rewritten.

 In summary, while the paper provides a valuable review of the literature on calf transportation, addressing the above points would enhance its clarity, focus, and overall contribution to the field of animal welfare during transit.

Round 2

Reviewer 3 Report (New Reviewer)

Comments and Suggestions for Authors

the paper improved a lot after the revisions

This manuscript is a resubmission of an earlier submission. The following is a list of the peer review reports and author responses from that submission.

Round 1

Reviewer 1 Report

Comments and Suggestions for Authors

Current knowledge on the transportation of weaned and unweaned calves by road

Transport is known to be one of the most stressful events for productive animals, although is an almost mandatory process for them. Several factors can alter the welfare during mobilization, particularly in calves since not always are considered different from cattle, and by their size, metabolism, physiology, and requirements, they may differ. The present article extensively describes these factors. However, in some parts, they are only described, and a deeper analysis or discussion seems to be missing. I left some comments below.

Simple summary and abstract: Please, clearly state the objective of the study.

Line 28: Apart from these factors, the type of road and even the driving skill may affect the welfare of animals.

Line 38: Please, include “animal welfare” as a keyword, to improve the visibility of the article.

Line 46: Correct the citation style (also on line 70).

Introduction: I suggest adding to this section other information such as the lack of non-invasive indicators to assess the physiological state of animals during and after transport. Also, in lines 48-49 it is mentioned that transport has negative welfare consequences. It would be adequate to include studies analyzing this factor and briefly explain its findings to understand the importance of this routine practice. These articles may be useful: https://doi.org/10.1016/j.prevetmed.2018.09.023 

 https://doi.org/10.1080/1828051X.2022.2038038

 https://doi.org/10.1017/S1751731110001989

Lines 49-51: I recommend rewriting the objective of the article and state that it analyzed the scientific literature available regarding truck transportation by road in weaned and un-weaned calves.

Line 54-73: Please, consider adding an introductory paragraph describing how a stressor is associated with cortisol and catecholamines secretion, and then mentioning that they can increase during loading or after transport. Since this review is focused on calves, it may be worth mentioning if the physiological response between calves and adult animals is different. Also, when mentioning physiological parameters such as heart rate, it would be appropriate to include the values (e.g., how much it increased) and mention that heart rate, respiratory rate, cortisol, and other biomarkers are not exclusively related to stress and response to many other physiological and pathological conditions. Therefore, behavior (the next paragraph after these lines) and a multimodal approach of both physiological and behavioral data are recommended.

Line 76: To mention the behavioral changes that can be observed in calves, I think it needs to be previously stated the association of stress and the main behavioral responses observed in calves. For example, after recognizing a stressor, what neural pathways are activated that trigger a behavioral consequence?

Line 77: Please, include the differences in the frequency of social interactions. If the aggression increased/decreased, or if animals showed more/less sexual behavior. The same applies to line 79 about exploratory behaviors (please, state which ones are considered exploratory behaviors); and to line 87. When mentioning that there was a change, it would be useful to clearly state if it increased/decreased and how much, so the reader can understand the difference before and after transport (e.g., lines 94, 97, 101, 103, 123, 134, etc.).

Line 86: Please, add the close parenthesis before reference 26.

Line 92: Please, add a reference

Line 100: This idea requires further discussion. For example, it can be mentioned that due to an increase in blood pressure, phosphorus efflux increase, causing renal damage to the glomerulus.

Line 108: Please, describe in which animals and for how many hours they were mobilized.

Line 120: This comment is in general throughout the manuscript since some paragraphs need more information than what is written. For example, when stating that bruises increase linearly with stocking density, it would be appropriate to include studies and examples saying that a stocking density above the recommended values caused 56% more bruising in the hindlimbs of animals. In this way, you are enriching the text and showing examples of what happens. I mention only this sentence, but the manuscript needs further analysis in other sections as well.

Line 124: Please, mention if this effect can reduce shelf life. Also, the authors could discuss the conditions that promote DFD meat or the biochemical process that occurs in animals before slaughtering when exposed to stress. These articles may help:   https://doi.org/10.1080/23144599.2020.1821574

https://doi.org/10.32854/agrop.v13i12.1927

Line 141 and 147: Please, revise the guide for authors for in-text citing style.

Line 203: Please, insert a reference.

Figure 1: I think colostrum intake needs to be mentioned somewhere inside the figure. Maybe between “birth” and “weaning”, since the acquisition of passive immunity is through colostrum intake, an event that is essential in the first hours of life for calves. Also, consider moving the figure after lines 223-235, since the figure is cited first in this paragraph.

Line 261: Please, specify the recommended stocking density for calves, with its respective reference.

Figure 2: The thermoneutral zone mentioned in lines 346-347 could be inserted inside the figure or described in the footnotes.

Line 348: I suggest discussing a little more about thermal stress in animals since it is one of the main factors that can affect animals during transport. Some of the current approaches recommended to evaluate this effect could also be mentioned. Please, revise  https://doi.org/10.1590/rbz4720160414 / DOI: 10.5923/j.zoology.20120204.03 / https://doi.org/10.1016/j.meatsci.2019.108025 / https://doi.org/10.3390/ani11123472 

Line 367: Please, further describe the findings from Anderson and Horder, not only mentioning the existence of the score, but if it has been used in calves, the results, other studies using the same score system, etc.  DOI: 10.1111/asj.13151 / Cattle transport: Historical, research, and future perspectives J. C. Swanson and J. Morrow-Tesch/ https://doi.org/10.2527/jas2001.79E-SupplE166x / DOI: 10.1002/vms3.2

Line 414: To improve the societal concerns, these articles may be helpful: https://doi.org/10.31893/jabb.22032 / https://doi:10.3390/ani8010004 / https://doi:10.3390/ani10030385 

Line 414: Before the conclusions, maybe the authors could add some paragraphs about recommendations when moving calves, according to the limitations found in the literature search.

Reviewer 2 Report

Comments and Suggestions for Authors

Dear authors,

You have chosen an important topic, however your entire manuscript needs thorough re-writing:

Since this manuscript is a review, I think you need to include some methodological considerations in the introduction, i.e. you should describe how you decided which papers to include and which not to include. For instance, I think it is a mistake not to include the work of Francesca Marcato.

The title of the manuscript mentions “weaned” and “unweaned calves” which makes me – the reader – expect a comparison of current knowledge about transport of calves of different age groups.  Nevertheless, in the text you often just refer to “calves” without mentioned their age. It is quite possible that the age of the calves is not mentioned in the individual studies you have included but then you need to address this lack of information.

Much of you text refers to studies done on adult cattle – I miss a discussion of the limitations of our current knowledge and that you reflect on which of these studies that are relevant for calves, so that it is clear for the reader what we know and do not know regarding calves and transport.

Below I have given you some comments, however it is important that you understand that in order to improve your manuscript sufficiently it is not enough to address these few comment – you need to take a critical look at the whole manuscript.

Line 26: You mention unweaned calves and adult cattle – what about the weaned calves from the title?

Lines 81-84: This is good. Here you make a comparison between weaned and unweaned. Same goes for lines 232-234 where you reflect on the differences between age groups.

Line 223: Help your reader and write to which age the “immune gap” refers.

Line 277-278: What is the difference between “related to balance” and “better calf stability”?

Figure 2. It needs to be clearly stated in the legend that this is not your own figure but an EFSA figure.

Lines 362-371: This is an example of a piece of text where you don’t mention age groups at all. Are these finding at all relevant for calves? You need to reflect on this.

Lines 377-379: Where is the reference? Is it no. 94? Rewrite to make this clearer.

Line 388-391: Animal welfare does not have one clear scientific definition, rather several suggestions exist. Rewrite to make this clearer.

Line 416-427: I suggest you rewrite your entire conclusion when you have rewritten the rest of the text. I am not interested in what EFSA thinks, I want to know what you conclude based on the studies you have included in your review. 

Reviewer 3 Report

Comments and Suggestions for Authors

Important revision from the point of view of collecting information on the transport of calves. As there are still few studies with the transport of live animals, articles like this one can serve as a parameter for others.

Reviewer 4 Report

Comments and Suggestions for Authors

General comments:

This is a nice review and plea for better conditions and handling of cattle before and during transportation. The title suggests that this manuscript deals with unweaned and weaned calves. However, at many points adult cattle are included.

I miss a chapter about the truck conditions. What is the effect of age of the vehicle? Level of maintenance? Furthermore, in a number of countries there is a mandatory course for the drivers and handlers. So what is the effect of driver education and experience (the latter just briefly mentioned). Also the road conditions are of major importance, this is also just briefly mentioned.

The temperature should be in the comfort zone of the calves, but how can this be achieved?

What is the ideal bedding material in a truck?

In the manuscript at many points the ‘optimal conditions’ are mentioned. But what these are, should be mentioned in the Conclusions.

Throughout the text there are several omissions in punctuation and a few spelling errors.

Specific comments:

Line 30: also water deprivation

Line 49: delete ‘can’. Transportation is always stressful.

Line 51: Transport by ship or rail is always preceded by transport by road.

Line 111: What components of the diet are important in this respect?

Chapter 2.1.3 and 2.1.4 can be combined.

Line 213-219: The content is here contradictory, so please explain a bit more.

Line 223:Unweaned calves are typically transported between 2-3 weeks in many countries.

Line 288: There is limited data. However, nothing is mentioned here. Please provide that limited data.

Line 365-367: There are many more studies that investigated the number of bruises, inflicted during transport, and the causal factors.

Line 389: Please give a source for the definition.